# Genetic Analysis of QTL for Resistance to Maize Lethal Necrosis in Multiple Mapping Populations

**DOI:** 10.3390/genes11010032

**Published:** 2019-12-26

**Authors:** Luka A. O. Awata, Yoseph Beyene, Manje Gowda, Suresh L. M., McDonald B. Jumbo, Pangirayi Tongoona, Eric Danquah, Beatrice E. Ifie, Philip W. Marchelo-Dragga, Michael Olsen, Veronica Ogugo, Stephen Mugo, Boddupalli M. Prasanna

**Affiliations:** 1Directorate of Research, Ministry of Agriculture and Food Security, Ministries Complex, Parliament Road, P.O. Box 33, Juba, South Sudan; lawata@wacci.ug.edu.gh; 2International Maize and Wheat Improvement Center (CIMMYT), World Agroforestry Centre (ICRAF), United Nations Avenue, Gigiri. P.O. Box 1041–00621, Nairobi, Kenya; Y.Beyene@cgiar.org (Y.B.); l.m.suresh@cgiar.org (S.L.M.); b.jumbo@cgiar.org (M.B.J.); M.Olsen@cgiar.org (M.O.); V.Ogugo@cgiar.org (V.O.); s.mugo@cgiar.org (S.M.); b.m.prasanna@cgiar.org (B.M.P.); 3West Africa Centre for Crop Improvement (WACCI), College of Basic and Applied Sciences, University of Ghana, PMB 30, Legon, Ghana; ptongoona@wacci.ug.edu.gh (P.T.); edanquah@wacci.ug.edu.gh (E.D.); bifie@wacci.ug.edu.gh (B.E.I.); 4Department of Agricultural Sciences, College of Natural Resources and Environmental Studies, University of Juba, P.O. Box 82, Juba, South Sudan; lukatwok11@gmail.com

**Keywords:** multiple population, linkage mapping, JLAM, QTL, validation, genomic prediction, maize lethal necrosis

## Abstract

Maize lethal necrosis (MLN) occurs when maize chlorotic mottle virus (MCMV) and sugarcane mosaic virus (SCMV) co-infect maize plant. Yield loss of up to 100% can be experienced under severe infections. Identification and validation of genomic regions and their flanking markers can facilitate marker assisted breeding for resistance to MLN. To understand the status of previously identified quantitative trait loci (QTL)in diverse genetic background, F_3_ progenies derived from seven bi-parental populations were genotyped using 500 selected kompetitive allele specific PCR (KASP) SNPs. The F_3_ progenies were evaluated under artificial MLN inoculation for three seasons. Phenotypic analyses revealed significant variability (*P* ≤ 0.01) among genotypes for responses to MLN infections, with high heritability estimates (0.62 to 0.82) for MLN disease severity and AUDPC values. Linkage mapping and joint linkage association mapping revealed at least seven major QTL (*qMLN3_130* and *qMLN3_142*, *qMLN5_190* and *qMLN5_202*, *qMLN6_85* and *qMLN6_157*
*qMLN8_10* and *qMLN9_142*) spread across the 7-biparetal populations, for resistance to MLN infections and were consistent with those reported previously. The seven QTL appeared to be stable across genetic backgrounds and across environments. Therefore, these QTL could be useful for marker assisted breeding for resistance to MLN.

## 1. Introduction

Maize lethal necrosis (MLN) is a major disease in sub-Saharan Africa (SSA) caused by co-infections of maize chlorotic mottle virus (MCMV) and sugarcane mosaic virus (SCMV) [1]. MCMV can able to interact with any member of the Potyviridae family to cause lethal necrosis in maize [2]. Yield loss due to MLN can reach up to 100% under severe infection and MLN favorable environments [1,3]. Breeding for host resistance to MLN is the most effective means of preventing yield losses in farmer’s fields. Application of molecular markers could enhance breeding for resistance to MLN. Although markers are widely used in breeding for crop improvement including maize, the tools are inconsequential unless the linked markers or quantitative trait loci (QTL) are tested for their effectiveness and reproducibility in different genetic backgrounds. QTL validation adds weight to assess the effectiveness of alleles and their linked markers.

QTL mapping approaches are one of the popular genomic tools to dissect the genetic architecture of complex traits [4]. The presence of QTL conferring resistance to several viral diseases in maize has been investigated in numerous linkage and linkage disequilibrium mapping studies [4,5,6,7,8]. QTL mapping or linkage mapping is known for high QTL detection power. Joint linkage association mapping (JLAM) based on different segregating biparental populations is known to provide both the high QTL detection power and high mapping resolution [9,10]. Application of both linkage mapping and JLAM in multiple bi-parental populations is useful to validate earlier findings and to detect possible new sources of resistance for MLN.

Several studies were conducted to validate QTL effects on traits of economic importance in different crops including maize [11,12,13,14,15]. Sukruth et al [16] validated the markers associated with late leaf spot and rust in groundnut using recombinant inbred line (RIL) population and two backcross populations. Zhou et al [17] validated two major QTLs (LEN-3H and LEN-4H) for kernel length in wild barley using a biparental population derived from Fleet (*Hordeum vulgare* L.) and Awcs276. The authors reported significant association between the two QTL and kernel length. Gowda et al [7] employed four biparental maize populations and adopted linkage mapping and joint linkage mapping options to discover and confirm QTL associated with resistance to MLN. They consistently confirmed in two of the populations that three major QTLs were localized on chromosomes 3, 6, and 9. For effective use of trait linked markers and resources, validation of discovered QTL with large expected impact is crucial, as there are too many QTLs to validate and validation population development and assessment is expensive [18,19,20]. It is a pre-requisite though that for QTL to be effectively used in crop improvement, it should be confirmed that their effects remain consistent across populations and environments [12,16,21]. 

QTL validation study greatly contributes towards increased resolutions of some of the target QTLs as well as complementarity to previous findings either from genome wide association studies (GWAS) or JLAM or any other approaches. The CIMMYT Global Maize Program has recently identified a number of QTL across maize chromosomes which are associated with resistance to MLN in multiple mapping populations [7,8]. The validation of these QTL using seven different F_3_ mapping populations would provide better understanding of QTLs associated with resistance to MLN and justify their application for marker assisted breeding towards improvement of maize lines for resistance to MLN. Further this study also helps to find new source of resistance which might not have been reported in earlier studies. This study was aimed to: (i) evaluate seven different F_3_ populations for their responses to MLN under artificial inoculation; (ii) conduct individual population-based QTL mapping; (iii) apply JLAM with three biometric models and compare with linkage mapping results to identify stable and/or unique QTL; and (iv) assess the potential of genomic prediction for MLN resistance within biparental populations with low marker density.

## 2. Materials and Methods

### 2.1. Plant Materials

Seven elite maize parental lines with contrasting response to MLN developed by CIMMYT through pedigree and DH breeding schemes were used in this study. Parents CKDHL120918, CML494, CKLTI0227, and CKDHL312 were tolerant to MLN; CML543 and CKDHL221 were moderately tolerant, and CKDHL0089 was susceptible to MLN. These materials were also known for tolerance to various biotic and abiotic stresses with good agronomic performances. The seven bi-parental populations (F_3_ pop1–CKDHL120918 × ML494, F_3_ pop2–CML543 × CML494, F_3_ pop3–CKDHL120918 × CML543, F_3_ pop4–CKLTI0227 × CKDHL120918, F_3_ pop5–CKDHL0089 × CKDHL120918, F_3_ pop6–CKDHL0221 × CKDHL120312 and F_3_ pop7–CKDHL0089 × CML494) were used for linkage mapping and JLAM. To develop F_3_ populations, crossing blocks were established in the nursery in Kiboko, Kenya (37°75′ E; 2°15′ S; 975 m a.s.l.; of 530 mm/year of rain fall and temperature ranges from 14.3 to 35.1 °C) during the 2016/2017 cropping season. Seven bi-parental crosses were made among the seven elite parental lines. Single cross (F_1_) seeds were grown and F_1_ plants were selfed. About 300 to 350 F_2_ plants from each population were randomly selfed and F_3_ seeds were harvested.

### 2.2. Phenotypic Evaluation

At least 306 F_3_ families from each population with their seven parental lines and six commercial checks were evaluated to determine their response to MLN under artificial inoculation in field. Experiments were conducted for three seasons (April 2017, April 2018, and October 2018) in confined MLN facility in Naivasha (36°26′ E; 0°43′ S; 1896 m a.s.l.; 677 mm/year of rain fall and temperature ranges from 12 to 29 °C), hereafter seasons are referred as environments. Trials were planted using alpha lattice design in a 1-row plot of 3.0 m long, with spacing of 0.75 m between rows and 0.25 m between plants. Two seeds were planted per hill and thinned to one plant per hill 3 weeks after germination, making a total of 13 plants per row. Standard agronomic practices were adopted. 

MLN inoculum was obtained from separate, sealed greenhouses maintained in Naivasha for each of SCMV and MCMV [7]. Maintenance of virus stocks in the greenhouses was described earlier [22]. In brief, MLN infected leaf tissues were collected from the field and ground in grinding buffer solution at 1:10 dilution ratio (10 mM potassium-phosphate, pH 7.0) [1,22]. The resulting sap extract was centrifuged at 12,000 rpm for 2 min. The sap was decanted and celite powder was added at the rate of 0.02 g/mL. To propagate the viruses in the greenhouses, a susceptible maize hybrid (H614) was grown and infected by rubbing the sap on the leaves at two to four leaf stages. A separate, sealed greenhouse was the maintained for each virus as stock for further use.

Three weeks before inoculation of the experimental materials, enzyme linked immunosorbent assay (ELISA) test was conducted on leaf samples, randomly collected from infected plants in the greenhouses, to determine the presence and purity of the MCMV and SCMV [7]. Separate extracts from the SCMV and MCMV infected plants was prepared and the two extracts were then mixed to form MLN inoculum at the ratio of 4 parts of SCMV to 1 part MCMV (weight/weight). Two inoculations were applied at the 4th and 5th week after planting. In order to keep uniform disease pressure across experiments, a motorized, backpack mist blower (Solo 423 Mist Blower, 12 L capacity) with an open nozzle (2-inches diameter) was used to inoculate the experimental plants at a high inoculum delivery pressure of 10 kg/cm. Drip irrigation was used to provide water and fertilizer. All other agronomic practices relating to maize production were followed according to standard procedures for field practices. Spreader rows of susceptible maize hybrid (H614) were planted along the experiment to enhance disease spread and intensity [16,23].

MLN Disease severity (MLN-DS) score started 10 days after the 2nd inoculations and was recorded four times at 10 days interval using standardized qualitative scale of 1 to 9, where 1 = resistant, clean, no symptoms; 2 = fine or no chlorotic specks, but vigorous plants; 3 = mild chlorotic streaks on emerging leaves; 4 = moderate chlorotic streaks on emerging new leaves; 5 = chlorotic streaks and mottling throughout plants; 6 = intense chlorotic mottling throughout plants, necrosis on leaf margins; 7 = excessive chlorotic mottling, mosaic and leaf necrosis, at times dead heart symptoms; 8 = excessive chlorotic mottling, leaf necrosis, dead heart and premature death of plants; and 9 = susceptible (complete plant necrosis and dead plants) [7,8]. After analyzing MLN-DS for each time score, we chose the third score (40 days post-inoculation) for further analysis because of its higher heritability and full expression of disease symptoms. The area under the disease progress curve (AUDPC) was calculated for each plot using SAS 9.4 (SAS Institute Inc., 2015, Cary, NC, USA) so as to understand the trend of development of MLN severity across the score intervals.

### 2.3. Phenotypic Data Analysis

All quantitative genetic parameters were estimated based on the performance of the 2142 F_3_ lines. To check the quality of the data, first, test for normality of distribution of error terms for MLN-DS and AUDPC was determined using the Shapiro-Wilk test [24]. Secondly, analyses of description statistics (mean, range, skewness and kurtosis) and correlation among phenotypic traits were performed using META-R [25]. We assumed that each environment was a representative of a replication. Therefore, statistical analysis of phenotypic data was conducted using linear mixed model:Y_ijk_ = μ + g_i_ + l_j_ + b_kj_ + ε_ijk,_(1) where Y_ijk_ was the disease severity of the ith genotype at the jth environment in the kth incomplete block, μ was an intercept term, g_i_ was the genetic effect of the ith genotype, l_j_ was the effect of the jth environment, b_kj_ was the effect of the kth incomplete block at the jth environment, and ε_ijk_ was the error term confounding with the genotype-by-environment interaction effect. To determine variance components by the restricted maximum likelihood (REML) method, both block and genotype effects were treated as fixed. Significance of variance component estimates was tested by model comparison with likelihood ratio tests where the halved *P*-values were used as approximations [26]. Heritability (*H*^2^) on an entry-mean basis was estimated as the ratio of genotypic to phenotypic variance. The phenotypic variance comprises genotypic variance and the masking GxE interaction variances divided by the number of environments. Further, the mixed linear model (MLN) established in META-R software (http://hdl.handle.net/11529/10201) was adopted and the best linear unbiased predictor (BLUP) and best linear unbiased estimator (BLUE) for each genotype across environments were generated. The BLUPs were used in linkage mapping and joint linkage association mapping analyses whereas BLUEs were used in genomic prediction studies.

### 2.4. DNA Extraction, Genotyping, Linkage Map Construction and QTL Analysis 

In this study, SNP markers were first screened on parental lines and 500 markers, which showed polymorphism between at least two of the seven parents were selected and used to genotype all the populations. Leaf samples were collected from 306 individuals per population 2 to 3 weeks after emergence, based on CIMMYT laboratory protocols [27]. The leaf samples were sent to LGC Genomics (https://www.biosearchtech.com/services/genotyping-services, Herts, UK) and were genotyped using 500 SNPs. Genotypic data obtained from LGC was subjected to quality check and SNPs were called and filtered using TASSEL version 5.0 software [28]. 

Polymorphic markers for each population were selected. Segregation of each SNP was verified for deviation from classic Mendelian inheritance using *x*^2^-test and SNPs that significantly deviated were discarded [29,30]. Linkage maps were created for the individual populations using the MAP function established in QTL IciMapping v 4.1 software [24,31]. Linkage groups were identified using Group command based on logarithm of odds (LOD) score of 3.0, and recombination rate were converted into centimorgans (cM) using Kosambi mapping function [32]. Ordering of the markers was conducted using the “ordering” instruction with the nnTwoOpt algorithm. Adjustment of the map order was done according to the sum of adjacent recombination frequencies (SARF) and sorted in the “rippling” instruction with a window size of 5 as the amplitude. Instruction generated from the map was used to draw and visualized the map using Map Chart software [33].

The BLUPs obtained from across environments for MLN-DS and AUDPC for each population were subjected to inclusive composite interval mapping (ICIM) analysis so as to determine the QTL linkage. The ICIM is an effective two-step statistical approach that allows separation of co-factor selection from interval mapping process, in order to control the background effects and improve mapping of QTL with additive effects [31]. A LOD threshold of 3.0 with a scanning step of 1 cM were used to declare significant QTL [21,34]. Stepwise regression was adopted to determine the percentages of phenotypic variance explained (*R*^2^) by individual QTL and additive effects at LOD peaks. QTL nomenclature [35] was adopted to nominate QTL conferring MLN resistance, where a two or three letter abbreviation of trait, followed by the chromosome number on which the QTL is found and the marker position to distinguish multiple QTLs were employed. The percentages of phenotypic variance explained (% PVE) by individual QTL, and additive and dominant effects at LOD peaks were generated. Sources of favorable alleles were determined depending on signs of the QTL additive effects [7,36]. For each F_3_ population, estimated additive (a) and dominance (d) effects for each QTL were used to calculate the ratio of dominance level (|*d*/*a*|). This ratio was used to classify the nature of QTL as follow [37]: additive (A; 0 ≤ |*d*/*a*| ≤ 0.2); partially dominant (PD; 0.2 ≤ |*d*/*a*| ≤ 0.8); dominant (D; 0.8 < |*d*/*a*| ≤ 1.2) and over dominant (OD; |*d*/*a*| > 1.2).

Based on the SNP markers shared by different populations, an integrated map was built by using IciMapping software [31]. In brief, SNPs overlapped across genetic maps were selected as anchor markers and used to integrate corresponding linkage groups on individual linkage maps. The marker order and marker positions were calculated after calculating the order and the relative position (within each genetic map) of the anchored markers, followed by integrating of all the detected markers into one map. Then all QTL identified from the seven populations were projected onto the integrated map based on their confidence interval. 

### 2.5. Joint Linkage Association Mapping (JLAM)

The JLAM method is a combination of linkage mapping and association mapping approaches [38,39]. Here, BLUPs obtained from across seven bi-parental populations and about 420 SNPs with missing values of <5% and minor allele frequency >0.05 were considered for JLAM study. We employed three biometric models to elucidate the QTL-trait relationships [7]. For each model, first, co-factors were selected using stepwise multiple linear regression based on the Schwarz Bayesian Criterion [40], following the Proc GLM SELECT model of SAS 9.2 [41]. Secondly, *P*-value and *F*-test were determined based on the full model (with QTL effects) and the reduced model (without QTL effects), followed by QTL scan using R software (version 3.5.3). The approach for Model A involved the trait as a function of the co-factor + marker. Model B incorporated population effect as an additional factor to adjust for the population structure so that trait = population + co-factor + marker; and Model C involved nesting of the co-factor and marker within population such that trait = pop + co-factor (pop) + marker (pop) [10,42]. Significance of association between QTL and MLN resistance was determined at *P* < 0.05 following Bonferroni-Holm procedure [43]. The adjusted total phenotypic variance explained (*R*^2^) values for the detected QTL were generated when the significant QTL were simultaneously fitted in linear model. Further principal component analysis (PCA) of all F_3_ lines was carried out using TASSEL version 5.2 [28], and the CurlyWhirly version 1.15 software [44] was used to construct PCA biplot for the population structure.

### 2.6. Genomic Prediction

Due to its advantages over either phenotypic and marker assisted selection, genomic prediction (GP) is becoming a popular approach for breeding, especially for complex traits [45,46]. The technique allows for the selection of superior genotypes without conducting phenotypic evaluation, if a subset of the population has genotypic and phenotypic data. Here, we performed GP analyses using ridge-regression BLUP (RR-BLUP) with five-fold cross-validation. Uniformly distributed, polymorphic, SNPs from each population showing minor allele frequency of less than 5% were used.

Genotypes were sampled from each population and used to constitute both training set and testing set. The sampling of the sets was repeated 100 times [7]. The predictive ability of the GP was computed by dividing the correlation between genomic estimated breeding values (GEBVs) and the observed phenotypic values by the square root of the heritability estimates obtained from the respective populations [46].

## 3. Results

### 3.1. Response of Parents and F_3_ Populations to MLN Infections

The response of F_3_ lines for MLN-DS and AUDPC showed continuous distribution, ranging from highly resistant or tolerant to completely susceptible in each individual population as well as across populations (Figure 1).

The normality tests indicated that the means were positively and moderately skewed with values ranging from 0.65 to 0.77. Kurtosis values were platykurtotic with maximum of 2.07 for AUDPC. High *W*-test values were observed for all the traits and ranged from 0.95 to 0.97 (data not shown). Individual means across populations ranged from 4.40 to 5.11 with a mean of 4.70 for MLN-DS, and the AUDPC mean values for individual populations ranged from 125.6 to 134.4 with an across population mean of 132.4. The magnitude of genotypic variance for MLN-DS was lowest (0.17) in F_3_ pop 2 (CML543 × CML494) and highest (0.69) in F_3_ pop 4 (CKLTI0227 × CKDHL120918). Genotypic variances were highly significant (*P* < 0.01) in all seven F_3_ populations and across populations for both MLN-DS and AUDPC values. Moderate to high broad–sense heritability estimates of 0.62 to 0.79 were found for MLN-DS and AUDPC values, indicative of high-quality phenotypic data for further genetic analysis (Table 1).

Three-dimensional principal component analysis (PCA) of the seven bi-prenatal populations are shown in Figure 2. The results identified five distinct groups including CML543 × CML494, CKDHL120918 × CML543, CKDHL0089 × CKDHL120918, CKDHL0221 × CKDHL120312, and CKDHL0089 × CML494. The distinctiveness of the five populations implies their diversity in their genetic backgrounds. However, populations CKDHL120918 × CML494 and CKLTI0227 × CKDHL120918 were overlapping and CKDHL221 × CKDHO120312 is distinct from other populations. This might indicate their relatedness and distinctness in genetic compositions. 

### 3.2. Molecular Analyses

#### 3.2.1. Linkage Group

Genetic linkage groups consisting of 10 maize chromosomes were constructed for each of the seven F_3_ populations. Marker density on each map varied among the seven populations. F_3_ pop 4 carried the largest number of polymorphic markers with 298 SNPs and a total length of 1550.07 cM at an average density of 5.20 cM between the markers. F_3_ pop 1 had only 112 polymorphic SNPs spanning a total length of 1223.97 cM with a mean spacing of 10.93 cM between markers. Similarly, variation was detected in number of SNPs per linkage group (LG), with marker density per LG ranging from 40 with a mean spacing of 5.71 for LG7 to 153 and an average of 21.86 on LG1. Total linkage map consisted of 389 SNPs with a total length of 2007.85 cM and an average marker interval length of 5.16 cM (Appendix A).

#### 3.2.2. QTL Mapping

QTL analyses across environments revealed 60 and 58 significant QTLs for MLN-DS and AUDPC values for the seven populations, respectively (Table 2), with different sizes of QTL effects. Generally, the study revealed that positive alleles for resistance to MLN were donated either by male or female parent in a cross in each population. Some QTL identified in F_3_ pop 6 and 7 showed large additive effects. QTL detected in the other F_3_ populations predominantly showed dominant to over-dominant effects. Majority of the QTLs was detected in F_3_ pop4 (CKLTI0227 × CKDHL120918), whereby 7% of the QTL showed additive effects for MLN-DS and AUDPC values.

In F_3_ pop 1, for MLN-DS and AUDPC values, together 15 QTL were detected, which explained 30.08% and 34.72% of the total PVE, respectively. Most of the detected QTL were of small effects except *qMLN6_85*, which had AUDPC values explaining 21.6% of PVE. In F_3_ pop 2, seven and 10 QTLs were detected which together explained 43.5 and 49.8% of the total phenotypic variance for MLN-DS and AUDPC values, respectively. Among all the QTLs identified, only three QTLs are specific to AUDPC values. In F_3_ pop3, a total of eight QTLs were found for MLN-DS, explaining 47.9% of the total observed phenotypic variability. Among the QTL with major effects, *qMLN3_142* was detected with 28.8% and 11.1% of PVE for MLN-DS and AUDPC values, respectively. There were 19 and 10 QTLs detected which together explained 25.1 and 28.7% of the PVE for MLN-DS and AUDPC values, respectively in F_3_ pop4. All the detected QTLs were of small effects. For F_3_ pop5, seven QTL each were found for MLN-DS and AUDPC values which together explained 54.3% and 57.6% of the total phenotypic variance, respectively. QTL *qMLN3_142* was found for both MLN-DS and AUDPC values with major effects. F_3_ pop6 revealed five QTLs explaining 52.1% of the variation for MLN-DS. Whereas for AUDPC values, nine QTLs were found which explained 59.1% of the total phenotypic variance. QTL *qMLN3_130* consistently explained >25% of the phenotypic variance. F_3_ pop7 revealed a total of three and four QTL with 46.7% and 50.8% PVE for MLN-DS and AUDPC values, respectively.

Among several genomic regions identified with stable QTL for MLN resistance, QTL on chromosome 3 was highly consistent. We found two strongly associated SNPs (*PHM15449_10* at 125,077,922 bp and *PZA00920_1* at 142,821,031 bp) within this region and their allelic effects on MLN resistance in different populations were also prominent. The phenotypic values of the different allele classes of these two major-effect SNPs in five F_3_ populations and across populations for MLN-DS and AUDPC value were presented in Figure 3.

#### 3.2.3. Consensus Map Construction

Since all populations were genotyped with the same set of SNPs, we integrated the seven maps into a consensus linkage map based on the markers shared by populations (Figure 4). From the total of 118 QTLs (60 for MLN-DS and 58 for AUDPC), similar QTL detected for both traits were treated as single QTL, as a result 77 QTLs were mapped on the consensus map. Among all the QTLs detected from seven populations, 19 QTLs were consistently found in at least two populations; QTL *qMLN3_142* on chromosome 3 was consistently identified in six of the seven populations. Two QTLs (*qMLN1_265* and *qMLN6_157*) were consistently found in four populations, seven QTLs (*qMLN1_47*, *qMLN3_130*, *qMLN4_150*, *qMLN6_85*, *qMLN7_158*, *qMLN8_10*, and *qMLN10_9*) were repeatedly identified in three populations and nine QTLs (*qMLN3_146*, *qMLN4_30*, *qMLN5_42*, *qMLN5_160*, *qMLN5_190*, *qMLN5_202*, *qMLN9_109*, *qMLN9_142*, and *qMLN10_114*) were consistently found in two populations. These clustered QTLs from different populations were placed on the consensus map (Figure 4). 

#### 3.2.4. Joint Linkage Association Mapping (JLAM)

The JLAM analyses based on three biometric models together found 27 and 28 main effect QTL for MLN-DS and AUDPC values (Table 3). For MLN-DS, 15, 12 and 18 QTLs were detected, which together explained 34.4%, 27.3% and 29.1% of the total variation in model A, B and C, respectively. Among 27 QTLs, five QTLs were consistently detected in all three models for MLN-DS. For AUDPC values, for model A, B and C, we found 14, 12 and 22 QTL which together explained 33.6%, 29% and 39.8% of the total variance. Seven QTLs were consistently detected in all three models. Most of the QTL detected through JLAM showed minor effects except one QTL *qMLN3_148* which was also consistent across models as well as across traits. Genomic prediction (GP) was attempted to predict the genetic values in the seven F_3_ populations for MLN-DS and AUDPC performances. The mean prediction accuracy ranged from as low as 0.12 in F_3_ pop1 to 0.75 in F_3_ pop 7 for MLN-DS (Figure 5). The prediction accuracies were similar for AUDPC values with range of 0.13 in F_3_ pop1 to 0.75 in F_3_ pop5. 

## 4. Discussion

### 4.1. Response of Parents and F_3_ Populations to MLN Infections

A total of 2142 F_3_ genotypes and seven parental lines were evaluated for MLN response in an artificially inoculated MLN plots for three seasons (2017–2018). Average mean for the F_3_ populations was lower than parental average for MLN-DS. Superiority of the populations over parents could be due to dominance and over-dominance effects resulting from combinations of favorable alleles from different parents and therefore, had better heterosis for resistance to MLN. The concept of heterosis has been widely reported in maize [47,48,49]. Significant mid-parent and best-parent heterosis for resistance to MLN across locations were reported in Northern Tanzania [50]. The observed heterosis could also be partially explained by higher vigor in the progenies than the inbred parents. The inbred parents are not as vigorous as the heterozygous progenies and less able to cope with stresses of various kinds including pathogens.

The significant variability observed among the populations for resistance to MLN is possibly due to genetic effects and could be an indication of differences in genetic backgrounds of the genotypes for resistance to MLN (Table 1). Further, the study detected high heritability estimates (0.71 to 0.94) for resistance to MLN in all the seven populations. Previous investigations revealed moderate to high heritability estimates (0.34–0.89) for resistance to MLN [7,22,48,51]. The high heritability estimates detected in this study corroborate earlier reports and strongly suggest that resistance to MLN could be largely influenced by several genes with major effects. 

### 4.2. QTL Analyses

In this study, seven major QTL were identified for resistance to MLN, localized on chromosomes 3 (*qMLN3_130* and *qMLN3_142*), 5 (*qMLN5_190* and *qMLN5_202*), 6 (*qMLN6_85* and *qMLN6_157*), 8 (*qMLN8_10*) and 9 (*qMLN9_142*) and spread across the 7-biparetal populations with the percentage of PVE ranging from 10.54% to 44.50% (Table 2). The results indicate that a large portion of the phenotypic variation in MLN resistance can be explained by few QTL with major effects while the remaining portion is due to QTL with minor effects. Similar observations have been made on various quantitative traits [52], including MLN [7,8]. It was observed in the present study that some QTLs showed major effects in some populations and minor effects in other populations. This variability could be attributed to various factors and gene actions including QTL × QTL interactions or QTL × environment interactions which might impact the size of effect of any given QTL in any given biparental population. 

To better understand the above interactions and their cause of variability among the genotypes, it is important to consider each component of gene action (additive, dominance and epistasis) separately. Additive effect is realized when each allele is independently contributing to genetic variability. Dominance effect is a variation caused by interaction between alleles at a locus. Epistasis effect is observed when the PVE is due to interactions between alleles at different loci. In this study, it was observed that QTL had varying levels of additive and dominance effects for resistance to MLN (Table 2), suggesting the importance of both modes of gene action in conditioning resistance to the disease. Similarly, some QTL had smaller %PVE values but had larger additive or dominance effects compared to the QTL with larger %PVE. Generally, when epistasis is involved, QTL with moderate and major additive effects are much more affected and they tend to have reduced %PVE. Therefore, epistasis could also be one of the main causes in some QTL having larger additive effects but with reduced %PVE values for resistance to MLN. QTL detected in F_3_ pop 1, 2, 3, 4, and 5 predominantly showed dominant to over-dominant effects. The dominance effects observed indicates MLN resistance is possibly due to interactions between individual alleles at specific loci. Other factors may include recombination between QTL and markers, leading to change in number of expected resistant alleles in the progeny hence reduced penetrance in QTL effects [53,54]. For QTL to be useful in a breeding program, it is important to first carry out validation studies to confirm their repeatability in different genetic backgrounds and environments. In line with expectations, the major QTL observed across populations in this study such as qMLN3_130, qMLN3_142, qMLN6_85, qMLN5_190, etc., indicated their stability across different genetic backgrounds. 

It was observed that favorable alleles for each QTL were contributed by either resistant or susceptible parent, which is indicated here by sign of the additive effect of the QTL. Negative (−) additive effect was considered for resistant parent being the source of favorable allele for resistance to MLN. Similarly, positive (+) additive effect implied that the susceptible parent was the donor of favorable allele for the observed phenotypic variability. This suggests the importance of major QTL in reduction of MLN infections and their stability across different genetic backgrounds [54]. Earlier studies [7,8] also detected contributions of favorable alleles from both resistant and/or susceptible parents. 

At least seven major QTLs localized across chromosomes and across populations with significantly larger %PVE were detected with total %PVE ranging from 25.13 to 59.07. Three of these QTLs (*qMLN3_130*, *qMLN6_85* and *qMLN5_190*) showed effects (%PVE), chromosome positions and confidence intervals (CI) comparable to the previous study [7]. For example, for *qMLN3_130*, the current CI is 125 to 169 Mb, whereas in the previous study CI was 125–130 Mb; for *qMLN6_85* CI is 84–91 Mb and from previous study CI was 85–96 Mb; and for *qMLN5_190* CI is 23–191 Mb and previous CI was 190–191 Mb. The larger CI observed in this study, compared to earlier study, could be due to few SNPs used in this study, however the results confirms the stability of the QTL in specific genomic regions. Another major QTL (*qMLN6_157*) detected on chromosome 6 in the current study was also found in the previous investigations [8]. This observation implies that three QTLs are stable across diverge genetic backgrounds, and hence can be useful in marker assisted breeding for resistance to MLN. Another major effect QTL *qMLN3_142* was detected in six out of the seven populations in this study, which also overlapped with QTL for MLN resistance in DH populations from the earlier study [8]. Two major QTLs (*qMLN5_202* and *qMLN8_10*) mapped across populations in this study, have not been reported previously. This implies the novelty of these genomic regions for MLN resistance. Similarly, it could suggest their specificity to the genetic backgrounds used in the current study. 

### 4.3. Joint Linkage Association Mapping (JLAM)

For JLAM, three biometric models were used to detect the maximum number of QTL linked to genes for MLN resistance. The principle of both models A and B resemble the composite interval mapping approach used in bi-parental populations [55] where co-factors were used to adjust for the population structure and background error within the segregating populations, leading to enhanced capacity to detect QTL. In both MLN-DS and AUDPC, we observed that >65% of QTL detected in model A and B overlapped with slightly higher %PVE in Model A (Table 3). This clearly indicates that control of population stratification by using co-factors alone was moderate, so population effect is also important. The slightly higher number of QTL detected by model A compared to model B could imply that model A was able to more effectively detect the variability among populations, and hence was able to reveal more QTL [10]. Model C with the nested marker effects, exploited the linkage disequilibrium (LD) within segregating populations and assumed population-specific QTL effects [10]. Model C was comparable with Model B in terms of the %PVE by these QTLs. This was well supported by all seven populations with relatively bigger population size (*n* = 306), where the nested model C is able to detect QTL with small to large effect size.

To evaluate the reliability of QTL detected in linkage mapping, we compared the identified 60 and 58 QTLs with 27 and 28 main effect QTLs detected through JLAM for MLN-DS and AUDPC values, respectively. Twenty-three QTLs are common across linkage mapping and JLAM for both the traits (Table 2 and Table 3). Comparison of QTL detected through JLAM and linkage mapping revealed five unique QTLs (*qMLN3_199*, *qMLN4_235*, *qMLN5_207*, *qMLN5_209*, and *qMLN6_13*) which were detected only through JLAM. QTLs common across methods helped to pinpoint the specific markers compared to QTLs detected with large CI through linkage mapping. Plausible gene candidates underlie some of the detected QTL regions with significant SNPs found in genes involved in plant defense system, like *PZA00447_8* associated with serine/threonine protein phosphatase, *PZA01313_2* linked to leucine-rich repeat protein and *PZA00726_8* linked to zinc ion binding function which has a role in plant defense responses (Appendix A). Among the unique QTL detected in JLAM, QTL on chromosome 6 seems to be overlapping with earlier reported major effect QTL for MLN resistance [7]. Major recessive QTL was reported in this region [56] in F_2_ populations. Similarly, with different association panels new QTL detected for resistance to SCMV in the same location on chromosome 6 [57]. Unique QTL on chromosomes 3 and 6 found in both model A and B but not in linkage mapping in this study indicates the variation across populations has helped to detect these QTL, which supports the use of multiple populations to find the novel source of resistance. On the contrary, the other three unique QTLs on chromosomes 4 and 5 were detected only in model C and were QTLs with population specific expression. 

We employed the same populations to evaluate GP by sampling both training and testing sets. Moderate to high accuracy of GP was observed for MLN-DS and AUDPC within populations. Low accuracy in F3 pop1 is also due to low number of polymorphic SNPs compared to other populations. Application of GP for selection for improvement of MLN resistance has been reported [7]. The previous investigations detected high prediction accuracy within the populations for both MLN-DS and AUDPC values. Resistance for MLN is complex and requires considerable resources and multiple selection cycles to identify resistant lines using phenotypic selection alone. Therefore, GP could provide an alternative practical approach for breeding for resistance to MLN because it accounts both major and minor effect QTL for accuracy, enhanced breeding gain (shortened breeding cycle) and reduced costs.

## 5. Conclusions

Seven major QTL were identified for resistance to MLN, localized on chromosomes 3 (*qMLN3_130* and *qMLN3_142*), 5 (*qMLN5_190* and *qMLN5_202*), 6 (*qMLN6_85* and *qMLN6_157*), 8 (*qMLN8_10*) and 9 (*qMLN9_142*) and spread across the 7-biparetal populations with a percentage of PVE ranging from 10.54% to 44.50%. Two SNPs on chromosome 3 (*PHM15449_10* at 125,077,922 bp and *PZA00920_1* at 142,821,031) were strongly associated with MLN resistance and their allelic effects in different populations were also prominent. Several QTL on chromosomes 1, 3, 5, and 6 shared similar CI with those reported previously. These QTL could be adopted for marker assisted breeding for improvement of maize against MLN infection. Additional three major QTL were not reported before further research is warranted to confirm their reproducibility and use in breeding for resistance to MLN. Both linkage mapping and JLAM can be incorporated to confirm earlier reported QTLs and discover new sources of resistance. Incorporation of GP can help to capture both large effect and small effect QTL to improve the level of MLN resistance as a result improved genetic gain. Twenty-three QTL were common across linkage mapping and JLAM however, five unique QTL (*qMLN3_199*, *qMLN4_235*, *qMLN5_207*, *qMLN5_209*, and *qMLN6_13*) were unique to JLAM with the QTL on chromosome 6 appearing to be overlapping with an earlier reported major effect QTL for SCMV resistance. Both linkage mapping and JLAM can be incorporated to confirm earlier reported QTL and discover new source of resistance. The incorporation of genomic selection can help to capture both large effect and small effect QTL to improve the level of MLN resistance.

## Figures and Tables

**Figure 1 genes-11-00032-f001:**
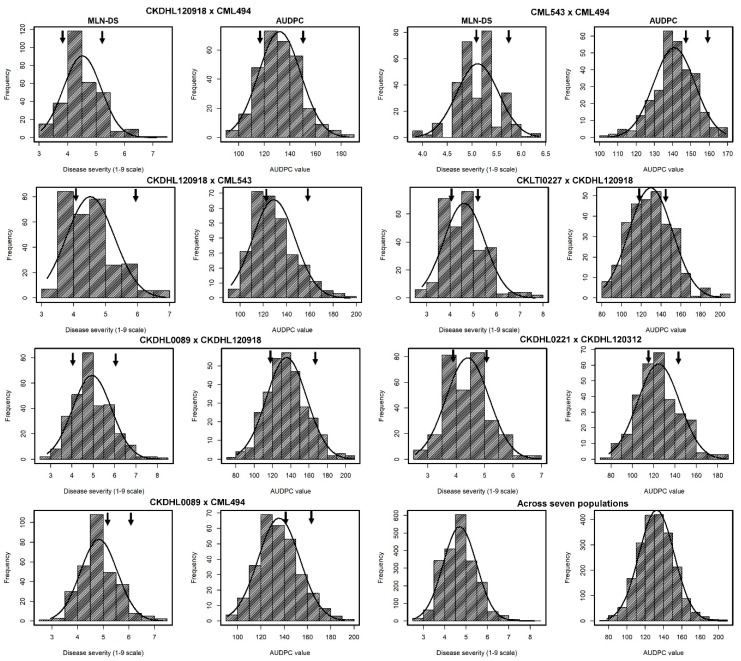
Phenotypic distribution of MLN disease severity (MLN-DS) and the area under disease progress curve (AUDPC) values recorded in seven individual and across F_3_ populations. Arrows indicate the performance of parents.

**Figure 2 genes-11-00032-f002:**
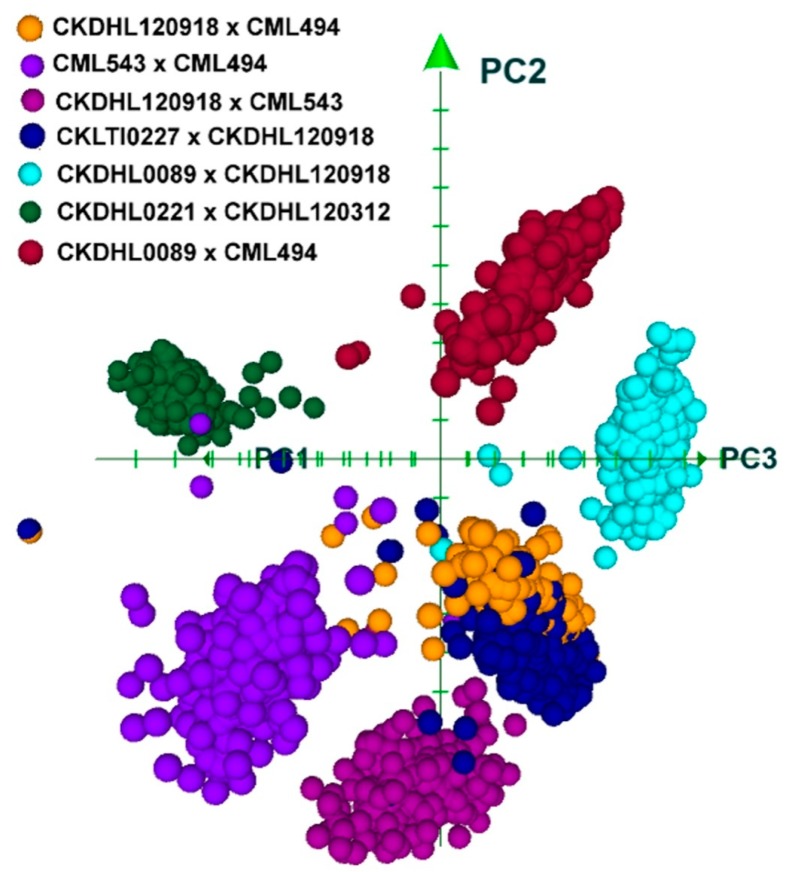
PCA (principal component analysis) biplot showing structures of the seven F_3_ populations.

**Figure 3 genes-11-00032-f003:**
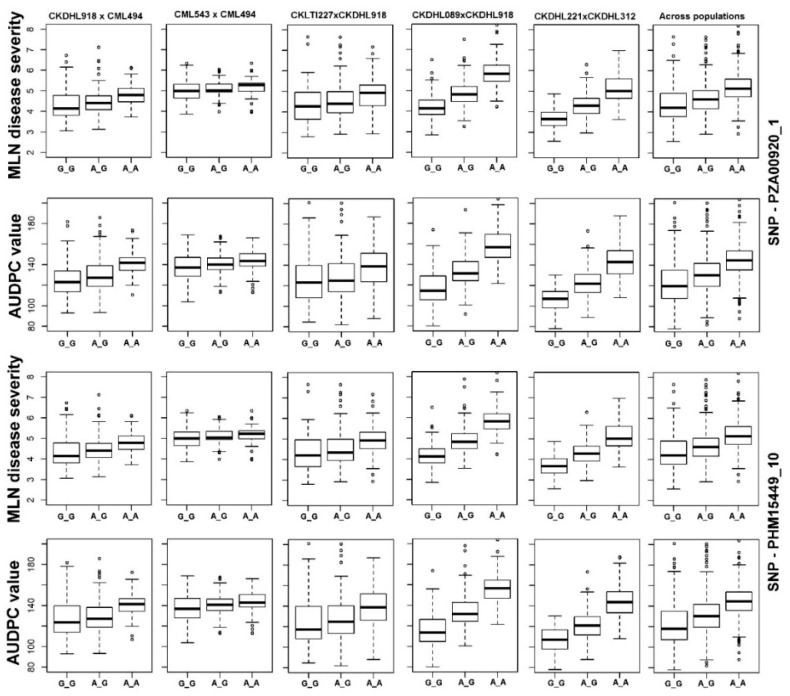
Box plots showing the phenotypic values of the different allele classes of two major-effect SNPs identified in five F_3_ populations and across populations for MLN-DS and AUDPC value. The SNP names, alleles and the specific F_3_ population where the effect is witnessed are mentioned above. The black horizontal lines in the middle of the boxes are the median values for the MLN-DS and AUDPC value in the respective allele classes. The vertical size of the boxes represents the inter-quantile range. The upper and lower whiskers represent the minimum and maximum values of data.

**Figure 4 genes-11-00032-f004:**
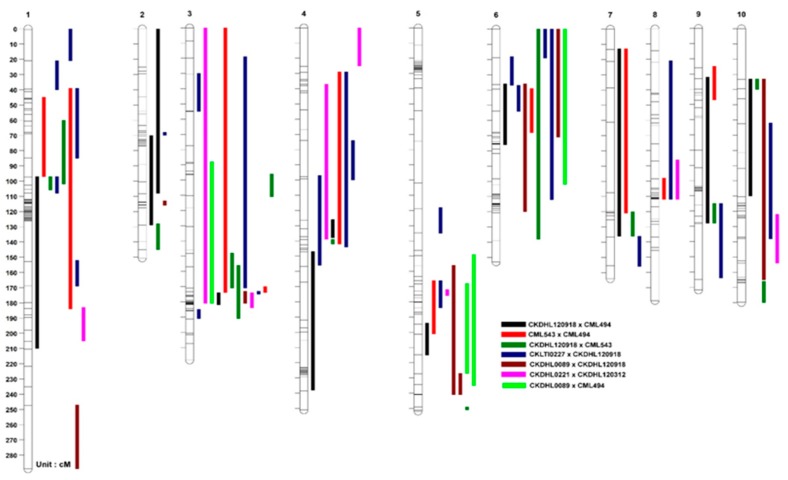
QTL for MLN resistance in the consensus linkage map of seven bi-parental populations. Different colors represent QTL detected from different populations.

**Figure 5 genes-11-00032-f005:**
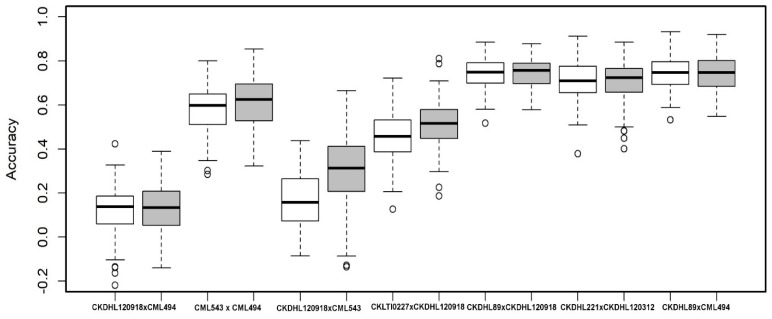
Box-whisker-plots for the accuracy of genomic predictions assessed by five-fold cross-validation within each population for MLN-DS (white box) and AUDPC values (grey box).

**Table 1 genes-11-00032-t001:** Analysis of variance for MLN disease severity (MLN-DS) and area under disease progress curve (AUDPC) values using seven segregating F_3_ populations evaluated for three seasons under MLN inoculated plots.

Trait	Mean (Range)	*σ^2^_G_*	*σ^2^_e_ **	*H* ^2^
CKDHL120918 × CML494 (F_3_ pop1)				
MLN-DS	4.52 (3.06–7.12)	0.41 **	0.44	0.74
AUDPC	131.9 (92.9–185.8)	272.20 **	230.73	0.78
CML543 × CML494 (F_3_ pop2)				
MLN-DS	5.11 (3.86–6.35)	0.17 **	0.32	0.62
AUDPC	140.8 (103.8–168.9)	141.05 **	177.68	0.70
CKDHL120918 × CML543 (F_3_ pop3)				
MLN-DS	4.52 (3.17–6.86)	0.46 **	0.50	0.73
AUDPC	126.2 (82.1–201.0)	282.50 **	280.87	0.75
CKLTI0227 × CKDHL120918 (F_3_ pop4)				
MLN-DS	4.60 (2.80–7.65)	0.69 **	0.57	0.79
AUDPC	129.1 (92.6–194.7)	467.78 **	299.86	0.82
CKDHL0089 × CKDHL120918 (F_3_ pop5)				
MLN-DS	4.94 (2.85–8.20)	0.65 **	0.72	0.73
AUDPC	133.9 (79.9–209.3)	411.12 **	379.39	0.76
CKDHL0221 × CKDHL120312 (F_3_ pop6)				
MLN-DS	4.40 (2.56–6.97)	0.50 **	0.49	0.75
AUDPC	125.6 (78.0–187.9)	374.25 **	245.35	0.82
CKDHL0089 × CML494 (F_3_ pop7)				
MLN-DS	4.82 (2.94–7.31)	0.44 **	0.51	0.72
AUDPC	134.4 (91.0–191.8)	300.75 **	260.51	0.78
Across seven populations				
MLN-DS	4.70 (2.56–8.20)	0.40 **	0.50	0.70
AUDPC	132.4 (78.0–209.3)	265.48 **	267.90	0.75

* and ** indicate significance at *P* < 0.05 and *P* < 0.01, respectively MLN-DS = MLN disease severity 42 dpi; AUDPC = area under disease progress curve; *σ^2^_G_* = genotypic variance; *σ^2^_e_** = error variance confounded with GxE variance; and *H*^2^ = broad sense heritability.

**Table 2 genes-11-00032-t002:** Quantitative trait loci (QTL) detected by integrated composite intervals mapping analysis for resistance to MLN in seven F_3_ populations evaluated in MLN inoculated plots over three cropping seasons.

Trait Name	QTL Name ^a^	Chr	Position (cM)	LOD	PVE (%)	Add	Dom	Nature of QTL	Total PVE (%)	Marker Name		Physical Position (bp)		Fav Parent
										Left M	Right M	Left M	Right M	
**CKDHL120918 × CML494**														
**MLN-DS**	*qMLN1_265*	1	183	3.99	2.58	−0.46	−0.26	PD	30.08	PZB00648_5	d8_3	17,595,139	265,199,938	CKDHL120918
	*qMLN2_156*	2	3	4.13	2.34	−0.04	1.09	OD		PZA01232_1	PHM3055_9	155,868,024	192,602,324	CKDHL120918
	*qMLN2_10*	2	122	3.16	0.69	0.07	−0.23	OD		PZA00620_3	PZA00365_2	10,429,405	1,221,385	CML494
	*qMLN3_142*	3	23	23.92	4.27	0.37	−0.21	PD		PZA00920_1	PHM15449_10	142,821,031	125,077,922	CML494
	*qMLN4_150*	4	90	4.41	2.53	−0.02	0.70	OD		S4_153520131	S4_149896839	153,520,131	149,896,839	CKDHL120918
	*qMLN5_177*	5	88	4.24	2.58	0.03	0.83	OD		PZA01410_1	S5_177634071	172,682,963	177,634,071	CML494
	*qMLN6_85*	6	78	14.11	3.98	0.34	−0.20	PD		PZB01009_1	PHM8909_12	84,664,840	91,883,155	CML494
	*qMLN6_85*	6	132	6.25	4.56	0.19	0.76	OD		PZB01009_1	PHM8909_12	84,664,840	918,83,155	CML494
	*qMLN7_158*	7	87	4.71	2.48	−0.07	1.11	OD		PHM7898_10	S7_157472460	161,993,743	157,472,460	CKDHL120918
	*qMLN9_142*	9	66	3.64	1.02	−0.11	−0.21	OD		PZB00221_3	PHM229_15	142,271,047	30,003,189	CKDHL120918
	*qMLN10_9*	10	4	3.82	2.43	0.08	0.84	OD		PZA00866_2	PHM5740_9	124,203,565	87,73,358	CML494
**AUDPC**	*qMLN2_10*	2	121	3.03	3.23	1.54	−6.16	OD	34.72	PZA00620_3	PZA00365_2	10,429,405	1,221,385	CML494
	*qMLN3_142*	3	23	12.68	8.92	5.50	−5.57	DO		PZA00920_1	PHM15449_10	142,821,031	125,077,922	CML494
	*qMLN4_143*	4	140	2.79	1.71	−2.78	1.27	PD		PHM1505_31	S4_9850443	143,162,745	9,850,443	CKDHL120918
	*qMLN6_85*	6	78	17.96	21.64	9.80	−5.38	PD		PZB01009_1	PHM8909_12	84,664,840	91,883,155	CML494
	*qMLN9_142*	9	68	4.79	6.25	−4.08	−5.16	OD		PZB00221_3	PHM229_15	142,271,047	30,003,189	CKDHL120918
**CML543 × CML494**														
**MLN-DS**	*qMLN1_47*	1	43	3.41	1.10	−0.07	0.04	PD	43.49	csu1138_4	S1_46411896	119,018,556	46,411,896	CML494
	*qMLN3_146*	3	14	11.79	4.08	−0.72	0.25	PD		S3_146966676	S3_146363360	146,966,676	146,363,360	CML494
	*qMLN4_30*	4	14	2.50	9.94	−0.22	0.52	OD		PZA02457_1	bt2_7	29,031,200	66,290,994	CML494
	*qMLN5_160*	5	3	3.52	1.22	−0.08	0.04	PD		S5_42297152	PZA01796_1	42,297,152	160,321,846	CML494
	*qMLN6_21*	6	4	5.03	1.64	0.09	−0.05	PD		S6_21007530	PZA03063_21	21,007,530	25,335,225	CML543
	*qMLN8_10*	8	84	3.42	12.37	−0.21	0.53	OD		S8_10001165	PZA02388_1	10,001,165	169,137	CML494
	*qMLN9_142*	9	3	31.21	12.72	0.70	−0.08	AD		PZA00832_1	PHM7916_4	147,131,097	132,762,904	CML543
**AUDPC**	*qMLN1_47*	1	42	6.14	1.64	−2.93	1.62	PD	49.84	csu1138_4	S1_46411896	119,018,556	46,411,896	CML494
	*qMLN1_265*	1	154	2.97	4.17	−7.58	9.26	OD		d8_3	umc128_2	265,199,938	227,602,208	CML494
	*qMLN3_146*	3	14	15.65	4.18	−24.01	7.57	PD		S3_146966676	S3_146363360	146,966,676	146,363,360	CML494
	*qMLN3_142*	3	107	3.77	1.17	2.73	0.84	PD		PZA00920_1	PZA02402_1	142,821,031	169,771,952	CML543
	*qMLN4_30*	4	14	3.88	7.99	−6.91	15.27	OD		PZA02457_1	bt2_7	29,031,200	66,290,994	CML494
	*qMLN5_160*	5	3	3.91	1.01	−2.48	0.94	PD		S5_42297152	PZA01796_1	42,297,152	160,321,846	CML494
	*qMLN6_21*	6	2	8.95	2.38	3.84	−0.71	AD		S6_21007530	PZA03063_21	21,007,530	25,335,225	CML543
	*qMLN7_158*	7	119	4.81	8.13	5.12	−18.38	OD		PHM7898_10	PHM1912_23	161,993,743	155,970,264	CML543
	*qMLN8_10*	8	85	3.05	5.28	−6.73	9.00	OD		S8_10001165	PZA02388_1	10,001,165	169,137	CML494
	*qMLN9_142*	9	3	34.18	10.54	21.95	−4.51	PD		PZA00832_1	PHM7916_4	147,131,097	132,762,904	CML543
**CKDHL120918 × CML543**														
**MLN-DS**	*qMLN1_265*	1	14	4.12	2.79	−0.04	−0.21	OD	47.88	d8_3	PHM14475_7	265,199,938	256,245,118	CKDHL120918
	*qMLN1_252*	1	28	3.31	3.08	−0.12	−0.15	OD		PZA02269_4	PHM4942_12	252,721,946	226,461,786	CKDHL120918
	*qMLN2_2*	2	50	4.30	3.45	−0.05	−0.23	OD		PHM13440_13	PZA00365_2	2,527,344	1,221,385	CKDHL120918
	*qMLN3_142*	3	73	33.33	28.84	0.46	−0.26	PD		S3_146966676	S3_68596995	146,966,676	68,596,995	CML543
	*qMLN4_143*	4	120	4.08	2.87	0.16	−0.04	PD		bt2_7	PHM1505_31	66,290,994	143,162,745	CML543
	*qMLN5_202*	5	112	2.89	4.43	−0.14	−0.19	OD		PZB00765_1	PHM5484_22	202,174,585	21,449,633	CKDHL120918
	*qMLN7_158*	7	1	2.62	3.15	−0.39	−0.44	DO		S7_157472460	S7_137455469	157,472,460	137,455,469	CKDHL120918
	*qMLN10_9*	10	81	6.77	5.10	−0.15	−0.22	OD		PHM5740_9	PZA01642_1	8,773,358	14,703,451	CKDHL120918
**AUDPC**	*qMLN1_265*	1	14	4.85	2.50	−1.32	−5.28	OD	52.80	d8_3	PHM14475_7	265,199,938	256,245,118	CKDHL120918
	*qMLN1_252*	1	28	3.44	2.40	−3.01	−3.52	DO		PZA02269_4	PHM4942_12	252,721,946	226,461,786	CKDHL120918
	*qMLN2_2*	2	50	4.03	2.46	−0.78	−5.45	OD		PHM13440_13	PZA00365_2	2,527,344	1,221,385	CKDHL120918
	*qMLN3_154*	3	3	2.93	5.23	10.02	−11.50	DO		S3_154250438	S3_150836832	154,250,438	150,836,832	CML543
	*qMLN3_142*	3	74	17.68	11.09	7.15	−6.24	DO		S3_146966676	S3_68596995	146,966,676	68,596,995	CML543
	*qMLN3_207*	3	109	2.73	2.02	−3.50	−0.77	PD		PZA00538_15	PZA01931_17	206,889,707	227,682,081	CKDHL120918
	*qMLN4_143*	4	120	6.03	3.30	4.58	−0.86	AD		bt2_7	PHM1505_31	66,290,994	143,162,745	CML543
	*qMLN5_202*	5	112	3.16	3.65	−3.50	−4.49	OD		PZB00765_1	PHM5484_22	202,174,585	21,449,633	CKDHL120918
	*qMLN6_157*	6	118	3.83	4.79	11.14	−8.68	PD		PZA01618_2	S6_156386857	129,927,781	156,386,857	CML543
	*qMLN7_158*	7	1	3.85	3.30	−10.90	−11.74	DO		S7_157472460	S7_137455469	157,472,460	137,455,469	CKDHL120918
	*qMLN9_109*	9	29	2.81	1.63	−3.06	−1.67	PD		PHM229_15	S9_109549230	30,003,189	109,549,230	CKDHL120918
	*qMLN10_41*	10	2	2.90	3.46	9.86	−10.22	DO		PHM4066_11	PHM1956_90	41,187,565	40,187,565	CML543
	*qMLN10_9*	10	82	7.45	4.61	−3.83	−5.70	OD		PHM5740_9	PZA01642_1	8,773,358	14,703,451	CKDHL120918
**CKLTI0227 × CKDHL120918**														
**MLN-DS**	*qMLN1_282*	1	99	3.21	1.55	0.69	−0.60	DO	25.13	PHM4752_14	PZA03020_8	298,874,066	282,044,048	CKLTI0227
	*qMLN1_265*	1	109	3.52	1.56	0.68	−0.63	DO		PZA03020_8	PZA01254_2	282,044,048	106,204,446	CKLTI0227
	*qMLN1_47*	1	139	4.79	1.38	0.76	−0.82	DO		S1_22744948	S1_87158930	22,744,948	87,158,930	CKLTI0227
	*qMLN1_27*	1	202	3.54	0.91	0.93	−0.94	DO		S1_15353866	S1_2693968	15,353,866	2,693,968	CKLTI0227
	*qMLN3_142*	3	37	4.44	1.45	0.73	−0.35	PD		S3_146966676	S3_55444954	146,966,676	55,444,954	CKLTI0227
	*qMLN3_130*	3	65	5.68	1.48	0.70	−0.82	DO		S3_92864540	S3_136165565	92,864,540	136,165,565	CKLTI0227
	*qMLN3_130*	3	121	5.32	1.49	−0.68	−1.08	OD		PHM2343_25	S3_154250438	27,981,649	154,250,438	CKDHL120918
	*qMLN4_30*	4	36	4.36	0.35	0.72	−0.05	AD		PZA02457_1	PZA00726_10	29,031,200	60,768,063	CKLTI0227
	*qMLN4_150*	4	96	4.29	1.59	0.59	−0.74	OD		S4_155296684	S4_9741874	155,296,684	9,741,874	CKLTI0227
	*qMLN5_160*	5	130	4.12	1.05	0.90	−0.83	DO		S5_154350617	S5_198716574	154,350,617	198,716,574	CKLTI0227
	*qMLN5_42*	5	161	3.49	0.25	−0.08	−1.02	OD		S5_42297152	PZA02164_16	42,297,152	112,179,855	CKDHL120918
	*qMLN6_157*	6	20	5.87	1.46	0.59	−1.12	OD		S6_156386857	PHM4748_16	156,386,857	158,540,019	CKLTI0227
	*qMLN6_85*	6	29	7.17	1.68	0.72	−0.34	PD		PHM4748_16	PZB01009_1	158,540,019	84,664,840	CKLTI0227
	*qMLN6_85*	6	83	8.86	1.79	0.81	−0.43	PD		PZB01009_1	S6_89823772	84,664,840	89,823,772	CKLTI0227
	*qMLN6_157*	6	116	4.73	1.10	0.07	1.37	OD		PHM2658_129	PZA01884_1	164,999,578	132,316,835	CKLTI0227
	*qMLN7_158*	7	145	3.56	1.08	0.86	−0.99	DO		S7_167230991	S7_127970539	167,230,991	127,970,539	CKLTI0227
	*qMLN8_10*	8	100	4.34	1.37	0.73	−0.90	OD		PZA02746_2	PZA02388_1	163,067,200	169,137	CKLTI0227
	*qMLN9_109*	9	9	3.19	1.64	−0.63	−0.57	DO		PZA00708_3	S9_109549230	147,381,231	109,549,230	CKDHL120918
	*qMLN10_114*	10	36	5.74	1.61	0.69	−0.83	OD		S10_113832226	PZA01001_2	113,832,226	146,538,889	CKLTI0227
**AUDPC**	*qMLN1_265*	1	163	6.85	1.35	24.93	−8.24	PD	28.69	d8_3	PZA02269_4	265,199,938	252,721,946	CKLTI0227
	*qMLN2_41*	2	2	12.20	2.31	40.41	−11.93	PD		PHM10404_8	PZA02450_1	40,967,991	47,575,949	CKLTI0227
	*qMLN3_142*	3	34	5.67	3.42	14.61	−8.98	PD		S3_146966676	S3_55444954	146,966,676	55,444,954	CKLTI0227
	*qMLN3_146*	3	113	10.79	1.98	−40.42	−5.50	AD		S3_146250249	S3_146026612	146,250,249	146,026,612	CKDHL120918
	*qMLN3_130*	3	125	3.49	3.81	12.36	−18.09	OD		PHM2343_25	S3_154250438	27,981,649	154,250,438	CKLTI0227
	*qMLN4_30*	4	77	3.29	2.44	19.01	−20.07	DO		S4_6601124	PZA02509_15	6,601,124	3,904,858	CKLTI0227
	*qMLN6_157*	6	32	8.84	3.77	16.09	−5.31	PD		PHM4748_16	PZB01009_1	158,540,019	84,664,840	CKLTI0227
	*qMLN9_147*	9	6	3.28	3.17	−15.11	−9.92	PD		S9_146012201	PZA00708_3	146,012,201	147,381,231	CKDHL120918
	*qMLN9_109*	9	21	3.19	2.66	18.29	−19.22	DO		PZA00708_3	S9_109549230	147,381,231	109,549,230	CKLTI0227
	*qMLN10_114*	10	36	3.54	3.20	16.73	−18.91	DO		S10_113832226	PZA01001_2	113,832,226	146,538,889	CKLTI0227
**CKDHL0089 × CKDHL120918**														
**MLN-DS**	*qMLN1_18*	1	4	4.42	7.15	−0.11	0.87	OD	54.34	S1_18838432	PZA00175_2	18,838,432	8,510,027	CKDHL120918
	*qMLN2_169*	2	168	4.25	6.61	−0.06	1.07	OD		PZA02727_1	PZA00515_10	227,921,381	169,265,278	CKDHL120918
	*qMLN3_142*	3	73	44.14	24.44	0.62	−0.10	AD		PZA00920_1	S3_133048570	142,821,031	133,048,570	CKDHL0089
	*qMLN5_190*	5	128	3.75	1.56	−0.14	−0.06	PD		S5_190675983	S5_201226926	190,675,983	201,226,926	CKDHL120918
	*qMLN6_157*	6	20	3.21	8.20	0.31	−0.41	OD		S6_156386857	PHM3466_69	156,386,857	167,148,384	CKDHL0089
	*qMLN6_85*	6	126	4.16	1.78	0.16	0.04	PD		PZA00942_2	PHM8909_12	102,566,000	91,883,155	CKDHL0089
	*qMLN10_9*	10	4	12.06	6.32	−0.28	−0.10	PD		PZA01313_2	PHM5740_9	3,598,262	8,773,358	CKDHL120918
**AUDPC**	*qMLN1_18*	1	10	3.69	8.65	−3.67	15.28	OD	57.62	S1_18838432	PZA00175_2	18,838,432	8,510,027	CKDHL120918
	*qMLN2_169*	2	166	3.18	6.28	−1.81	21.55	OD		PZA02727_1	PZA00515_10	227,921,381	169,265,278	CKDHL120918
	*qMLN3_142*	3	73	50.02	27.46	16.42	−2.21	AD		PZA00920_1	S3_133048570	142,821,031	133,048,570	CKDHL0089
	*qMLN5_190*	5	127	5.49	2.11	−4.27	−1.58	PD		PHM7908_25	S5_190675983	191,075,472	190,675,983	CKDHL120918
	*qMLN6_157*	6	21	2.96	8.12	7.34	−10.37	OD		S6_156386857	PHM3466_69	156,386,857	167,148,384	CKDHL0089
	*qMLN6_85*	6	133	5.03	1.96	4.23	0.36	AD		PHM8909_12	PZA00427_3	91,883,155	79,815,961	CKDHL0089
	*qMLN10_9*	10	3	11.23	5.20	−6.46	−2.44	PD		PZA01313_2	PHM5740_9	3,598,262	8,773,358	CKDHL120918
**CKDHL0221 × CKDHL120312**														
**MLN-DS**	*qMLN1_47*	1	71	5.32	3.74	−0.16	0.07	PD	52.13	PZA00447_8	S1_46411896	9,024,005	46,411,896	CKDHL120312
	*qMLN3_130*	3	56	44.83	44.51	0.56	−0.11	AD		PZA02402_1	PHM15449_10	169,771,952	125,077,922	CKDHL0221
	*qMLN4_150*	4	37	2.66	2.74	−0.13	−0.09	PD		PZA01187_1	PHM1505_31	177,666,738	143,162,745	CKDHL120312
	*qMLN4_7*	4	128	4.63	3.23	0.15	0.01	AD		S4_6544767	PHM16788_6	6,544,767	13,581,955	CKDHL0221
	*qMLN8_10*	8	45	4.26	3.12	0.15	0.03	AD		PHM5235_8	PZA00368_1	94,414,978	5,632,308	CKDHL0221
**AUDPC**	*qMLN1_47*	1	71	4.97	4.38	−4.05	1.41	PD	59.07	PZA00447_8	S1_46411896	9,024,005	46,411,896	CKDHL120312
	*qMLN1_47*	1	100	2.98	2.49	2.21	−2.90	OD		S1_46411896	PHM12323_17	46,411,896	53,357,797	CKDHL0221
	*qMLN3_130*	3	56	24.64	26.00	9.70	−2.76	PD		PZA02402_1	PHM15449_10	169,771,952	125,077,922	CKDHL0221
	*qMLN3_142*	3	63	13.19	12.71	6.76	−1.85	PD		PZA00279_2	PZA00920_1	52,804,070	142,821,031	CKDHL0221
	*qMLN4_150*	4	39	3.65	5.24	−4.27	−2.01	PD		PZA01187_1	PHM1505_31	177,666,738	143,162,745	CKDHL120312
	*qMLN4_7*	4	128	3.95	3.44	3.71	0.18	AD		S4_6544767	PHM16788_6	6,544,767	13,581,955	CKDHL0221
	*qMLN5_42*	5	34	2.63	2.36	−0.23	−4.17	OD		PHM16854_3	PZA00522_12	34,587,029	57,933,548	CKDHL120312
	*qMLN8_10*	8	45	5.06	4.70	4.16	−0.42	AD		PHM5235_8	PZA00368_1	94,414,978	5,632,308	CKDHL0221
	*qMLN10_114*	10	43	5.16	4.47	3.72	−2.52	PD		PZA00814_1	PHM1576_25	87,194,491	124,203,168	CKDHL0221
**CKDHL0089 × CML494**														
**MLN-DS**	*qMLN5_190*	5	89	4.40	5.48	−0.17	0.03	AD	46.74	PZA01427_1	PHM7908_25	23,135,578	191,075,472	CML494
	*qMLN5_202*	5	111	8.94	7.97	−0.20	−0.03	AD		S5_200938637	PHM563_9	200,938,637	204,993,639	CML494
	*qMLN6_157*	6	107	3.37	8.42	0.09	0.71	OD		S6_157568432	S6_156386857	157,568,432	156,386,857	CKDHL0089
**AUDPC**	*qMLN3_130*	3	1	3.02	6.64	0.75	−4.18	OD	50.87	PZA01447_1	S3_133048570	53,549,251	133,048,570	CKDHL0089
	*qMLN5_190*	5	89	5.14	15.91	−4.78	0.81	AD		PZA01427_1	PHM7908_25	23,135,578	191,075,472	CML494
	*qMLN5_202*	5	111	7.62	16.65	−4.81	−0.76	AD		S5_200938637	PHM563_9	20,0938,637	204,993,639	CML494
	*qMLN6_157*	6	108	3.20	14.95	2.34	16.81	OD		S6_157568432	S6_156386857	157,568,432	156,386,857	CKDHL0089

LOD = logarithm of odds; Add = additive effect; Dom = dominance effect; PVE = phenotypic variance explained; fav parent = parental genotype from where favorable allele for MLN resistance is contributing; ^a^ QTL name composed by the trait code followed by the chromosome number in which the QTL was mapped and a physical position of the QTL.

**Table 3 genes-11-00032-t003:** Joint linkage association mapping depicting allele substitution effects and total phenotypic variance explained (PVE) using segregating F_3_ progenies derived from seven bi-parental populations evaluated for three seasons under MLN inoculation.

Marker	QTL Name	Chr	Position (Mbp)	Model A		Model B		Model C	
				α-Effect	PVE (%)	α-Effect	PVE (%)	α-Effect	PVE (%)
PZA00447_8	*qMLN1_9*	1	9.02	−0.08	0.60	–	–	–	–
PHM5622_21	*qMLN1_184*	1	183.83	–	–	–	–	−0.12	0.40
S3_48493677	*qMLN3_48*	3	48.49	0.32	7.90	–	–	−0.02	0.40
S3_55444954	*qMLN3_55*	3	55.44	0.14	0.10	0.10	1.00	–	–
S3_68596995	*qMLN3_68*	3	68.60	–	–	−0.06	0.20	0.06	0.40
S3_92694873	*qMLN3_92*	3	92.69	−0.28	0.70	–	–	−0.02	1.80
S3_113429913	*qMLN3_113*	3	113.43	–	–	−0.38	1.00	–	–
PHM15449_10	*qMLN3_125*	3	125.08	0.10	3.30	0.11	0.80	0.16	2.20
S3_148291047	*qMLN3_148*	3	148.29	−0.72	10.20	−0.66	4.70	−0.16	1.40
S3_151342843	*qMLN3_151*	3	151.34	−0.23	3.20	−0.42	3.30	–	–
PHM2919_23	*qMLN3_199*	3	199.89	−0.12	0.40	−0.12	0.40	–	–
PZA00726_8	*qMLN4_60*	4	60.77	–	–	–	–	−0.04	0.70
S4_235381719	*qMLN4_235*	4	235.38	–	–	–	–	0.03	0.90
PHM565_31	*qMLN5_24*	5	24.24	–	–	−0.03	0.00	−0.47	0.30
S5_170023563	*qMLN5_170*	5	170.02	–	–	–	–	0.01	0.10
PHM7908_25	*qMLN5_191*	5	191.08	–	–	–	–	0.04	0.20
S5_196017729	*qMLN5_196*	5	196.02	−0.11	0.10	−0.03	0.10	−0.01	0.20
S5_202816906	*qMLN5_202*	5	202.82	–	–	–	–	−0.10	0.10
PHM563_9	*qMLN5_204*	5	204.99	–	–	–	–	−0.08	0.20
PZA03167_5	*qMLN5_207*	5	207.60	–	–	–	–	0.32	0.30
S5_209467974	*qMLN5_209*	5	209.47	–	–	–	–	−0.07	0.30
S6_13300385	*qMLN6_13*	6	13.30	0.19	1.90	0.20	1.10	–	–
S6_86475982	*qMLN6_86*	6	86.48	−0.27	1.00	–	–	–	–
S6_89823772	*qMLN6_90*	6	89.82	−0.24	2.70	−0.23	0.90	−0.22	2.50
PHM5235_8	*qMLN8_94*	8	94.41	0.15	0.50	–	–	–	–
PZA01313_2	*qMLN10_4*	10	3.60	0.11	1.30	0.18	2.90	0.10	0.80
PHM5740_9	*qMLN10_9*	10	8.77	0.09	0.50	–	–	–	–
Total PVE (%)					34.40		27.30		29.10
PZA00447_8	*qMLN1_9*	1	9.02	−1.63	0.20	–	–	–	–
PHM5622_21	*qMLN1_184*	1	183.83	–	–	–	–	−3.54	0.40
S3_48493677	*qMLN3_48*	3	48.49	6.63	6.80	–	–	–	–
S3_55444954	*qMLN3_55*	3	55.44	3.53	1.00	1.94	0.60	–	–
S3_68596995	*qMLN3_68*	3	68.60	–	–	−2.45	0.60	−1.55	2.30
PHM15449_10	*qMLN3_125*	3	125.08	–	–	3.01	1.00	3.65	3.00
S3_148291047	*qMLN3_148*	3	148.29	−20.14	10.20	−16.47	4.80	−3.86	1.50
S3_151342843	*qMLN3_151*	3	151.34	−12.06	4.90	−10.59	3.70	−4.72	1.80
PHM2919_23	*qMLN3_199*	3	199.89	−4.35	0.50	−4.04	0.80	−1.70	0.80
PZA00726_8	*qMLN4_60*	4	60.77	–	–	–	–	−1.64	0.70
S4_155378923	*qMLN4_155*	4	155.38	–	–	−8.03	1.00	–	–
S4_235381719	*qMLN4_235*	4	235.38	–	–	–	–	−10.74	0.80
S5_170164477	*qMLN5_170*	5	170.16	5.53	0.50	–	–	10.13	0.50
PHM7908_25	*qMLN5_191*	5	191.08	–	–	–	–	1.38	0.20
S5_196017729	*qMLN5_196*	5	196.02	−2.14	0.20	−1.13	0.40	−4.37	0.20
S5_202816906	*qMLN5_202*	5	202.82	–	–	–	–	–2.53	0.10
PHM563_9	*qMLN5_204*	5	204.99	–	–	–	–	–5.61	0.30
PZA03167_5	*qMLN5_207*	5	207.60	–	–	–	–	7.57	0.30
S5_209467974	*qMLN5_209*	5	209.47	–	–	–	–	2.00	0.40
S6_13300385	*qMLN6_13*	6	13.30	6.30	4.30	5.69	1.40	0.54	1.00
S6_86475982	*qMLN6_86*	6	86.48	–6.97	0.90	–	–	–	–
S6_89823772	*qMLN6_90*	6	89.82	–6.58	0.90	–6.14	1.00	–5.81	2.00
S8_74144408	*qMLN8_74*	8	74.14	–	–	–	–	3.86	0.30
PHM5235_8	*qMLN8_94*	8	94.41	4.41	1.50	–	–	–	–
S8_102533570	*qMLN8_102*	8	102.53	–	–	–	–	0.83	0.30
PZA01313_2	*qMLN10_4*	10	3.60	3.41	1.20	4.73	3.30	0.66	0.90
PHM5740_9	*qMLN10_9*	10	8.77	–	–	–	–	–5.51	0.30
PZA00866_2	*qMLN10_124*	10	124.20	1.71	0.60	1.44	0.40	1.79	0.90
Total PVE (%)					33.60		29.00		39.80

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
