# Peer review of "Genetic Analysis of QTL for Resistance to Maize Lethal Necrosis in Multiple Mapping Populations"

_genes, 2019, doi:10.3390/genes11010032_

Round 1

Reviewer 1 Report

Authors need to improve the introduction and discussion which is not scientifically sound.

The corrections that need to be done before publication of this manuscript are as below :    Author can site this study in the introduction somewhere (for eg. in line 37) Wamaitha, Mwathi Jane et al. “Metagenomic analysis of viruses associated with maize lethal necrosis in Kenya.” Virology journal vol. 15,1 90. 23 May. 2018, doi:10.1186/s12985-018-0999-2

Line 108 [1][21].
Line 464  Gustafson et al. (2018)    Line no.  143 : (Payne 2006). this ref. needs to be consistent.

Line 367  Karume et al 

There are many mistakes in the citations in the main text and in Reference list so the authors need to check those manually.

Otherwise, the study is scientifically sound and of agricultural importance. I have no concern after corrections of a few things.

Author Response

Comments and Suggestions for Authors

Comment 1

Authors need to improve the introduction and discussion which is not scientifically sound.

Response: Thanks for your suggestion. We modified the introduction and discussion section in the revised manuscript. Please see Line 45 to 67

Comment 2

The corrections that need to be done before publication of this manuscript are as below :    Author can site this study in the introduction somewhere (for eg. in line 37) Wamaitha, Mwathi Jane et al. “Metagenomic analysis of viruses associated with maize lethal necrosis in Kenya.” Virology journal vol. 15,1 90. 23 May. 2018, doi:10.1186/s12985-018-0999-2

Response: We included the suggested new reference in the revised manuscript

Comment 3

Line 108 [1][21].
Line 464  Gustafson et al. (2018)    Line no.  143 : (Payne 2006). this ref. needs to be consistent.

Response: Sorry for the mistakes related to reference. We modified them appropriately in the revised manuscript.

Comment 4 

Line 367  Karume et al 

 Response: Sorry for the mistakes related to reference. We modified this sentence and reference appropriately in the revised manuscript.

Comment 5

There are many mistakes in the citations in the main text and in Reference list so the authors need to check those manually.

Response: Thanks for pointing out this mistake related to references. We rechecked them and modified them appropriately wherever it is necessary in the revised manuscript.

Comment 6

Otherwise, the study is scientifically sound and of agricultural importance. I have no concern after corrections of a few things.

Response: Thanks for your constructive comments on this manuscript

Reviewer 2 Report

The authors of the manuscript „Genetic analysis of QTL for resistance to maize lethal necrosis in multiple mapping populations” present at QTL analysis for the maize lethal necrosis disease, which represents a devastating corn disease. To better understand the genetic variation underlying this disease, the authors studied seven bi-parental populations using 500 KASP SNPs. The aim was to validate previously found QTLs as well as identify novel associated genomic regions. Overall the major findings, such as the QTLs and their stability, of this study are well supported. I do find the methodology sound and the results statistically supported. I only observed a few points that should be addressed.

Larger points:

Previous studies (Ref 6, 7) also identified QTLs for MNL using GWAS. Those previous studies also highlighted potential target genes for some of the major QTLs. The authors found overlapping QTLs e.g. on chromosome 3. The study would provide much more novelty, if for those overlapping/stable QTLs, some information on the candidate genes would be provided. For example, if those candidate genes show differential expression between infection and control conditions. The results from figure 2 are not clear. The authors have previously shown that pop 2 and 4 (CML 543x CML 494) and (CKLT10227 x CKDHL120918) showed the largest variance in their genotypic response. However, the 3D PCA in figure2 shows that these two populations are the most overlapping. This indicates their close genetic relatedness. I think these two contrasting results of the largest variance but in PCA closest relatedness should be more clearly discussed. The authors identified two novel QTLs which were stable across the tested population not found in REF 6,7. The authors suggest that this is maybe due to the novelty of genetic backgrounds. Are the used markers polymorphic in the REF 6, 7 tested lines? If so due they also show an associated with MLN? Since some authors are shared between those previous studies, access those previously used populations may exist.

Minor points:

The authors mention in 246/247 that the magnitude genotypic variance was lowest for pop2 (CML 543x CML 494) with 0.10. However, if I understand Table 1 correctly that should be 0.17. Line 247 I feel the genetic crosses /populations should be specified instead of just pop2 and pop4. The results from figure 2 are not clear. The authors have previously shown that pop 2 and 4 (CML 543x CML 494) and (CKLT10227 x CKDHL120918) showed the largest variance in their genotypic response. However, the 3D PCA in figure2 shows that these two populations are the most overlapping. This indicates their close genetic relatedness. I think these two contrasting results of the largest variance but in PCA closest relatedness should be more clearly discussed. Line 260 spelling: “of the seven bi-parental populations are shown in Figure 2? Table 2: I think table 2 is too large and lacks focus for the main manuscript text. My suggestion would be to restrict it to the major QTLs with the additional QTLs shown in the supplemental data. Lines 407 -410 needs more clear explanation. The OTL should be specified and the stability explained based on the results. Line 401 spelling: are much more affected? Line 403: Given the size of Table2 which significant dominant effects mentioned here should be more clearly stated. Line 425 spelling: the results confirm; in a specific genomic region or in specific genomic regions? Line 428 spelling: genetic backgrounds? Line 432/433: “This implies the novelty of these genomic regions in manifestation of MLN resistance.” What do the authors mean with this statement?

Author Response

Comments and Suggestions for Authors

Comment 1

The authors of the manuscript „Genetic analysis of QTL for resistance to maize lethal necrosis in multiple mapping populations” present at QTL analysis for the maize lethal necrosis disease, which represents a devastating corn disease. To better understand the genetic variation underlying this disease, the authors studied seven bi-parental populations using 500 KASP SNPs. The aim was to validate previously found QTLs as well as identify novel associated genomic regions. Overall the major findings, such as the QTLs and their stability, of this study are well supported. I do find the methodology sound and the results statistically supported.

Response: Thank you for the positive comment

Comment 2

I only observed a few points that should be addressed. Larger points:

Previous studies (Ref 6, 7) also identified QTLs for MLN using GWAS. Those previous studies also highlighted potential target genes for some of the major QTLs. The authors found overlapping QTLs e.g. on chromosome 3. The study would provide much more novelty, if for those overlapping/stable QTLs, some information on the candidate genes would be provided. For example, if those candidate genes show differential expression between infection and control conditions.

Response: We really appreciate your comments to conduct the study on differential expression of genes which surely give more clarity on resistance mechanism involved in MLN. However our objectives are limited till to find the trait associated markers in diverse genetic back grounds and environments, so that we can use them in breeding. Sure we will consider your suggestion on differential gene expression studies in our future experiments.

Comment 3

The results from figure 2 are not clear.

Response: We provided PCA figures to have an idea on population structure across 7 populations which helps on drawing conclusions for JLAM based results. The Figure 2 was based on 3 PCs however in a picture it has been shown as 2D way, to show clear 3D picture we have to include the video of the PCA which is big file size, so we used the present figure. The results of the figure is summarized in results section in Line 261 to 266 in the revised manuscript.

Comment 4

The authors have previously shown that pop 2 and 4 (CML 543x CML 494) and (CKLT10227 x CKDHL120918) showed the largest variance in their genotypic response. However, the 3D PCA in figure2 shows that these two populations are the most overlapping. This indicates their close genetic relatedness. I think these two contrasting results of the largest variance but in PCA closest relatedness should be more clearly discussed.

Response: Thank you for your comment. We believe in the Figure 2, pop 1 and pop 4 are overlapping not pop 2 and 4. It also expected because both populations are sharing one common parent (CKDHL120918). Therefore we were not discussed about pop 2 and 4 in the text.

Comment 5

The authors identified two novel QTLs which were stable across the tested population not found in REF 6,7. The authors suggest that this is maybe due to the novelty of genetic backgrounds. Are the used markers polymorphic in the REF 6, 7 tested lines? If so due they also show an associated with MLN? Since some authors are shared between those previous studies, access those previously used populations may exist.

Response: Thank you for your suggestion. When we cross checked the SNPs with earlier MLN discovery results, it is clear that most of the SNPs used in the current study are selected from genomic regions where MLN major QTL were reported from earlier studies. However these markers not showed any association with MLN resistance even though they are located physically closer to the major effect QTL for MLN. Only in this study they showed their association with MLN under new genetic back ground, therefore we called them as new or novel QTL and expressed in the new genetic background.

Comment 6

Minor points:

The authors mention in 246/247 that the magnitude genotypic variance was lowest for pop2 (CML 543x CML 494) with 0.10. However, if I understand Table 1 correctly that should be 0.17.

Response: Sorry for the mistake, we corrected it in the revised manuscript.

Comment 7

Line 247 I feel the genetic crosses /populations should be specified instead of just pop2 and pop4.

Response: Thank you for your suggestion. We included the pedigrees in the revised manuscript. Please see Line 250

Comment 8

The results from figure 2 are not clear. The authors have previously shown that pop 2 and 4 (CML 543x CML 494) and (CKLT10227 x CKDHL120918) showed the largest variance in their genotypic response. However, the 3D PCA in figure2 shows that these two populations are the most overlapping. This indicates their close genetic relatedness. I think these two contrasting results of the largest variance but in PCA closest relatedness should be more clearly discussed.

Response: Thank you for your suggestion. When we check the Table 1, the magnitude of genetic variance is small for pop 2 with 0.17 and high for pop 4 with 0.69 for MLN-DS. In Figure 2, pop1 and pop 4 are overlapped and is expected since they are sharing one common parent but pop 2 and pop 4 are not overlapped. Therefore we not included any additional information about pop 2 and pop 4, though we mentioned about closeness of pop 1 and 4.

Comment 9

Line 260 spelling: “of the seven bi-parental populations are shown in Figure 2?

Response: Thank you showing the mistake and we corrected it in the revised manuscript.

Comment 10

Table 2: I think table 2 is too large and lacks focus for the main manuscript text. My suggestion would be to restrict it to the major QTLs with the additional QTLs shown in the supplemental data.

Response: Thank you for the suggestion. Initially we also thought about it, however this table has the core results for the whole study, therefore we believed that it should be part of the main text so we decided to keep as it is in the revised manuscript.

Comment 11

Lines 407 -410 needs more clear explanation. The OTL should be specified and the stability explained based on the results.

Response: Thank you for the suggestion and we modified the relevant discussion in the revised manuscript.

Comment 12

Line 401 spelling: are much more affected?

Response: Thank you showing the mistake and we corrected it in the revised manuscript.

Comment 13

Line 403: Given the size of Table2 which significant dominant effects mentioned here should be more clearly stated.

Response: Thank you for the suggestion and we included the relevant information in the revised manuscript, please see Line 404-407.

Comment 14

Line 425 spelling: the results confirm; in a specific genomic region or in specific genomic regions?

Response: it is for specific regions, we corrected it in the revised manuscript.

Comment 15

Line 428 spelling: genetic backgrounds?

Response: Thank you showing the mistake and we corrected it in the revised manuscript.

Comment 16

Line 432/433: “This implies the novelty of these genomic regions in manifestation of MLN resistance.” What do the authors mean with this statement?

Response: Thank you showing the mistake and we modified the specific sentence it in the revised manuscript.

Reviewer 3 Report

The authors report the analysis of a large QTL mapping study for Maize lethal necrosis resistance using seven F3 mapping populations. They find high heritability for this trait in all populations and a large number of QTL. Due to somewhat limited resolution within populations, it total they find a large portion of the genome is associated in at least one population, suggesting a large number of segregating small-effect QTL. In total, they find ~60 QTL, with several that explain >20% of the variation within populations, and many more with minor effects. The authors also employ joint association mapping across the populations, but this detected fewer QTL than the union of the single-population QTL mapping.    Overall, the experimental design, data, and analysis appear to be of high quality. I have a few comments about methods and presentation:   Major: I don’t understand the justification for the 3 JLAM methods. It seems to me only Model C is reasonable. Population clearly needs to be added as a factor given the differences in mean value across populations. And covariates also clearly should be nested within populations because background QTL will also vary across populations. A model like Model B that forces the marker effect to be the same across populations could be reasonable, but only if we could assume that the SNPs used to genotype the populations have the same phase with the causal locus in all populations which seems unlikely. I would recommend only including Model 3. I’m a bit surprised that JLAM has so much lower power than the within-population approaches. This suggests to me that the within-population approaches are using too relaxed significance threshold, or the JLAM threshold is too stringent. It appears significance thresholds are set at 3LOD for within-population, and Bonferroni-corrected p<0.05 for JLAM. One consideration: if the aim is a level=0.05 test within each population, because you’re combining results across 7 populations your chance of a single false positive will be ~30%. To correct, it would be better to use a level = 0.05/7 threshold within each population. I’m not sure what this would correspond to in LOD, but a better way to select thresholds is by permuting genotypes.   Minor: It would be useful to report statistics on the parents of the populations as well as the F3 individuals. 149: why was genotype fixed and not environment? For H2 you’d want genotype random and maybe l_j fixed I think 211: What R function was used for the QTL scan? Figure 2: 3D plots are hard to see. How about 2 (or 3) 2D plots (PC1 vs PC2, PC1 vs PC3, PC2 vs PC3)? Also, how much variation was explained by PCs 1-3? It could be that the overlapping populations were separated on PC4 which may not capture much less variation. 283: This sentence is unclear “fewer QTL identified … are associated with higher number of additive effect QTLs” 287: This paragraph is really just re-stating Table 2 and so is redundant. Can you summarize these results instead? Figure 3: It might be useful to also show the parental values on these plots for comparison, or include in Figure 1 Figure 4: Text is too small to read. Also, how were QTL confidence intervals estimated? I’m not sure what the genomic prediction adds to this story. It’s not really closely related to the goal (measuring the similarity of QTL effects across populations)

Author Response

Comments and Suggestions for Authors

Comment 1

The authors report the analysis of a large QTL mapping study for Maize lethal necrosis resistance using seven F3 mapping populations. They find high heritability for this trait in all populations and a large number of QTL. Due to somewhat limited resolution within populations, it total they find a large portion of the genome is associated in at least one population, suggesting a large number of segregating small-effect QTL. In total, they find ~60 QTL, with several that explain >20% of the variation within populations, and many more with minor effects. The authors also employ joint association mapping across the populations, but this detected fewer QTL than the union of the single-population QTL mapping.    Overall, the experimental design, data, and analysis appear to be of high quality.

Response: Thank you your constructive comments on our results.

Comment 2

I have a few comments about methods and presentation:   Major: I don’t understand the justification for the 3 JLAM methods. It seems to me only Model C is reasonable. Population clearly needs to be added as a factor given the differences in mean value across populations. And covariates also clearly should be nested within populations because background QTL will also vary across populations. A model like Model B that forces the marker effect to be the same across populations could be reasonable, but only if we could assume that the SNPs used to genotype the populations have the same phase with the causal locus in all populations which seems unlikely. I would recommend only including Model 3.

Response: Thank you for your suggestion. Before the analyses we were not sure about which model suits or performs better in our research populations, therefore we used all three models. Later, as you suggested, we agree with you that model C makes more sense to include and also to some extent model B. We included all model as an additional information for readers but for the conclusion we considered only model B and C.

Comment 3

 I’m a bit surprised that JLAM has so much lower power than the within-population approaches. This suggests to me that the within-population approaches are using too relaxed significance threshold, or the JLAM threshold is too stringent. It appears significance thresholds are set at 3LOD for within-population, and Bonferroni-corrected p<0.05 for JLAM. One consideration: if the aim is a level=0.05 test within each population, because you’re combining results across 7 populations your chance of a single false positive will be ~30%. To correct, it would be better to use a level = 0.05/7 threshold within each population. I’m not sure what this would correspond to in LOD, but a better way to select thresholds is by permuting genotypes.  

Response: Thank you for your valuable comments. Yes we agree with you that the stringency of significance threshold for JLAM is higher than QTL mapping. We also needed high stringency in JLAM because JLAM is acting as a final validation step to choose markers associated with MLN resistance, therefore we can test those SNPs in deployment. However we agree with your comment that using same level of threshold will give more appropriate comparison across methods, however, since we focused on selecting few SNPs with high consistency for deployment we used high stringent threshold for JLAM.  

Comment 4

Minor: It would be useful to report statistics on the parents of the populations as well as the F3 individuals.

Response: Thank you for your comment. Since the parents were already screened for polymorphism of the markers used, we thought presenting the parent per se would not be necessary.

Comment 5

149: why was genotype fixed and not environment? For H2 you’d want genotype random and maybe l_j fixed I think

Response: Thank you for your suggestion. In this study genotypes were considered fixed to estimate BLUEs and considered as random to estimate variance components and BLUPs. Environment is always fixed because of few number or df, here number of environments are 3.

Comment 6

211: What R function was used for the QTL scan?

Response: QTL scanning for Models- 1D scan with F test for every marker locus was used. Full versus reduced model with F test

Comment 7

Figure 2: 3D plots are hard to see. How about 2 (or 3) 2D plots (PC1 vs PC2, PC1 vs PC3, PC2 vs PC3)? Also, how much variation was explained by PCs 1-3? It could be that the overlapping populations were separated on PC4 which may not capture much less variation.

Response: As we explained in response to the previous comment, the PCA figure was included to give a picture of population structure across 7 populations which helps on drawing conclusions for JLAM based results. The Figure 2 was based on 3 PCs however in a picture it has been shown as 2D way, to show clear 3D picture we have to include the video of the PCA which is big file, so we used the present figure. The results of the figure are summarized in results section in Line 261 to 266 in the revised manuscript.

Comment 8

283: This sentence is unclear “fewer QTL identified … are associated with higher number of additive effect QTLs”

Response: Thank you for your comment.  We have corrected this in the revised manuscript please, see lines 285-285.

Comment 9

287: This paragraph is really just re-stating Table 2 and so is redundant. Can you summarize these results instead?

Response: Thank you for your comment.  We have corrected this in the revised manuscript. Please see Line 286-294

Comment 10

Figure 3: It might be useful to also show the parental values on these plots for comparison, or include in Figure 1 Figure 4: Text is too small to read.

Response: Thank you for your comment.  We have improved the figures 1 and 4 as for the suggestion in the revised manuscript.

Comment 11

Also, how were QTL confidence intervals estimated?

Response: Thank you for your comment. The confidence intervals were estimated using bootstrap method based on 95% confidence interval with a walking speed of 1.0 cM. This function is also inbuilt in ICIM mapping QTL software

 Comment 12

I’m not sure what the genomic prediction adds to this story. It’s not really closely related to the goal (measuring the similarity of QTL effects across populations)

Response: Thank you for your comment.  The analysis of genomic prediction was attempted to validate the relative importance of the method in breeding for resistance to MLN.